# A simple technique to improve performance of four-port wideband MIMO antenna using shorting pins

**Anh Tran-Tuan[1], Thao Hoang-Thi-Phuong[2], Yem Vu-Van[3], Hung Tran-Huy[1]***

**1** Faculty of Electrical and Electronic Engineering, PHENIKAA University, Yen Nghia, Ha Dong, Hanoi, Vietnam, **2** Faculty of Electronics and Telecommunications, Electric Power University, Hanoi, Vietnam, **3** School of Electrical and Electronic Engineering, Hanoi University of Science and Technology, Hanoi, Vietnam

* hung.tranhuy@phenikaa-uni.edu.vn

**Data availability statement:** All relevant data are within the paper.

## Abstract

This paper presents a four-port multiple-input multiple-output (MIMO) antenna with compact size, wideband operation, and high isolation characteristics. The wideband performance is achieved by generating two adjacent resonances, which are respectively produced by a half-wavelength slot and a metasurface (MS). Four MIMO elements are arranged in a 2 × 2 configuration with zero spacing between the MIMO elements to achieve the compact size feature. For mutual coupling reduction, the adjacent elements are positioned so that their polarizations are perpendicular. Meanwhile, the coupling between the opposite elements is suppressed with the aid of shorting pins. The final design has compact size of 1.08 $\lambda$ × 0.89 $\lambda$ × 0.07 $\lambda$ and center-to-center element spacing of 0.35 $\lambda$, where $\lambda$ is the free-space wavelength at 5.2 GHz. The measured operating bandwidth, in which the reflection and transmission coefficients are respectively smaller than −10 and −20 dB, is from 4.9 to 5.8 GHz. Across this band, the proposed design has peak broadside gain of 4.5 dBi and good MIMO diversity performance. Compared to other related works, the proposed design has advantages of wideband, high isolation, while keeping a compact size characteristic.

## Introduction

By using Multiple-Input Multiple-Output (MIMO) technology, modern wireless communication systems can significantly increase the channel capacity and the spectrum efficiency [1]. In the MIMO systems, when multiple antennas are positioned in a closely spaced arrangement, the high mutual coupling between the neighbouring MIMO elements will deteriorate the system performance. Consequently, mutual coupling reduction is one of the most inevitable tasks when designing MIMO antenna. Additionally, the demands for high-speed data transfer and multiple wireless services integrating into a single device also led to the necessity of wideband operation antenna. In this paper, the authors focus on designing MIMO antenna, which simultaneously has high isolation and wideband operation. It is also noted that only MIMO designs with low-profile configuration are mentioned in this paper.

**Funding:** This work was supported by the Vietnam National Foundation for Science and Technology Development (NAFOSTED) under Grant number 102.04-2023.28.

**Competing interests:** The authors have declared that no competing interests exist.

For the sake of coupling suppression, various approaches have been considered in open literature. Generally, the decoupling structures can be categorized into three main types. The first attempt is to create an orthogonal operating mode on the non-excited element with the aid of a near-field resonator [2,3] or structures that can convert polarization [4–6]. An alternative method is to introduce extra structures working as band-reject filters between the radiators [7–9], in which the additional circuit elements will directly block the coupling from the excited elements to the adjacent ones. For the remaining technique, extremely low mutual coupling could be obtained by generating new coupling paths that can counter the original coupling with the assistance of loading aperture [10–12] or parasitic elements [13,14]. Although high isolation can be achieved, the above-mentioned decoupling structures always require extra space in either horizontal or vertical directions, leading to large inter-element spacing or high-profile configuration.

Regarding the operating bandwidth (BW), monopole or slot antennas are widely used for wide operation purposes [15–21]. This antenna type features by omni-directional beam, which is suitable for wide coverage communication. For long-range communication distance, uni-directional radiation pattern is more preferred. In [22–24], three different dielectric resonator (DR) antennas are proposed. In [25–27], wideband operation with microstrip patch structure is also achieved. Alternatively, metasurface (MS) is also another effective solution to improve the operating BW of the slot/patch antenna [28,29]. However, these structures suffer from critical disadvantages of high profile and/or large element distance due to the space required for decoupling network. Furthermore, the limited number of MIMO elements is also another drawback of several designs.

This paper presents a four-port MIMO antenna with compact size, wideband operation, and high isolation characteristics. The wideband operation is based on the combination of slot and MS. Meanwhile, mutual coupling is suppressed with the aid of eight shorting pins. Note that the pins are positioned inside the MIMO elements. Thus, no additional space is required for the decoupling structure, which contributes to reducing the distance between the MIMO elements.

## Wideband MIMO element

Recently, MS has been widely used to design wideband antenna with low-profile configuration. This method is applied in this paper by incorporating MS with the primary radiation from a slot. Fig 1 shows the geometrical configuration of two different wideband antennas. Both antennas consist of a 50-$\Omega$ microstrip line as feeding part, a half-wavelength slot, and a $2 \times 2$-unit cell MS. The difference between these designs is the orientation of the slot. For Design-1, it is arranged in $x$-direction. Meanwhile, the slot of Design-2 is in diagonal direction of the MS.

The reflection coefficient $|S_{11}|$ comparison between Design-1 and Design-2 is shown in Fig 2. Similar performance is achieved for both designs. The simulated –10 dB impedance BW is from 4.4 to 6.0 GHz, equivalent to about 30.7%. Here, wideband operation is achieved by combining two adjacent resonances. The first resonance is produced by the half-wavelength slot, which is excited by the feeding microstrip line. In addition, the MS layer plays the role of a parasitic element. It is coupled with the slot to generate additional resonance. Tuning the size of the unit cell, $w$, the higher resonance will be significantly affected. Meanwhile, the lower resonance is strongly influenced by the length of the slot, $l_s$.

## Mutual coupling reduction for adjacent MIMO elements

In this Section, Design-1 and Design-2 are utilized to produce 2-port MIMO antenna. Fig 3 shows the configurations of different MIMO antennas. The MIMO-1 consists of two Design-1

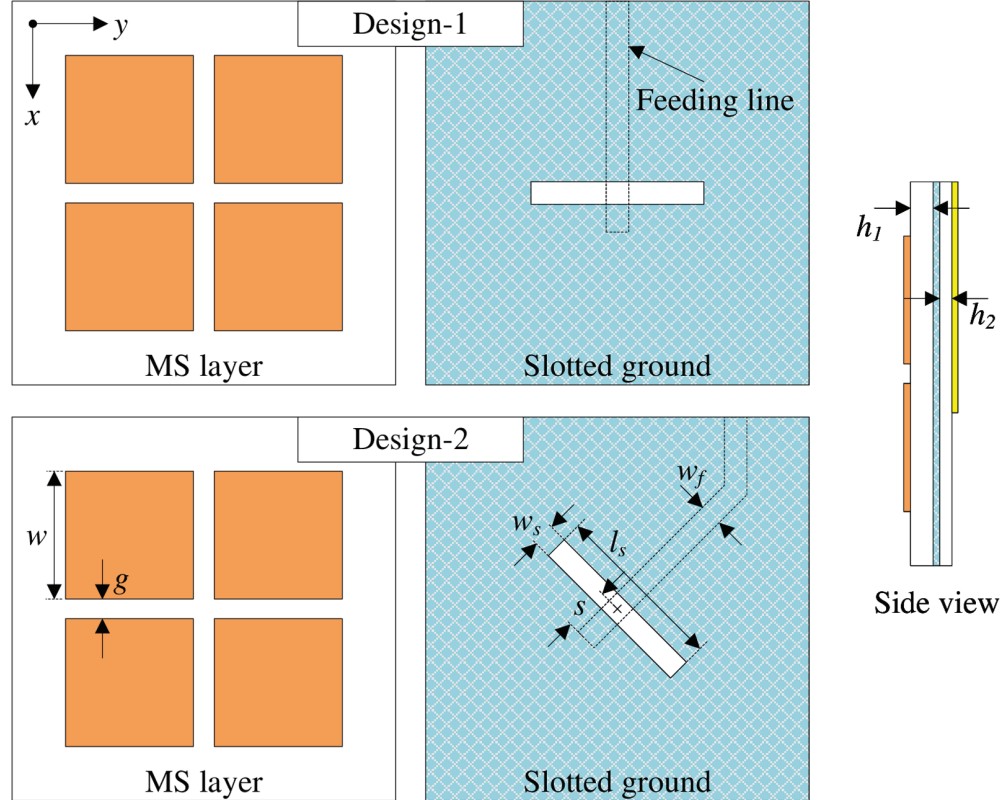

**Fig 1. Geometry of Design-1 and Design-2.** The optimized values of Design-2 are as follows: $w = 8.4$, $g = 1.4$, $l_s = 11.2$, $w_s = 1.3$, $w_f = 1.5$, $s = 3.0$ (unit: mm).

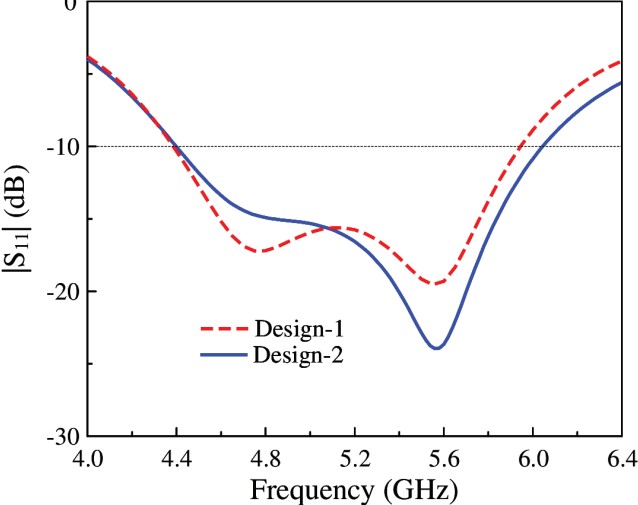

**Fig 2. Simulated $|S_{11}|$ of Design-1 and Design-2.**

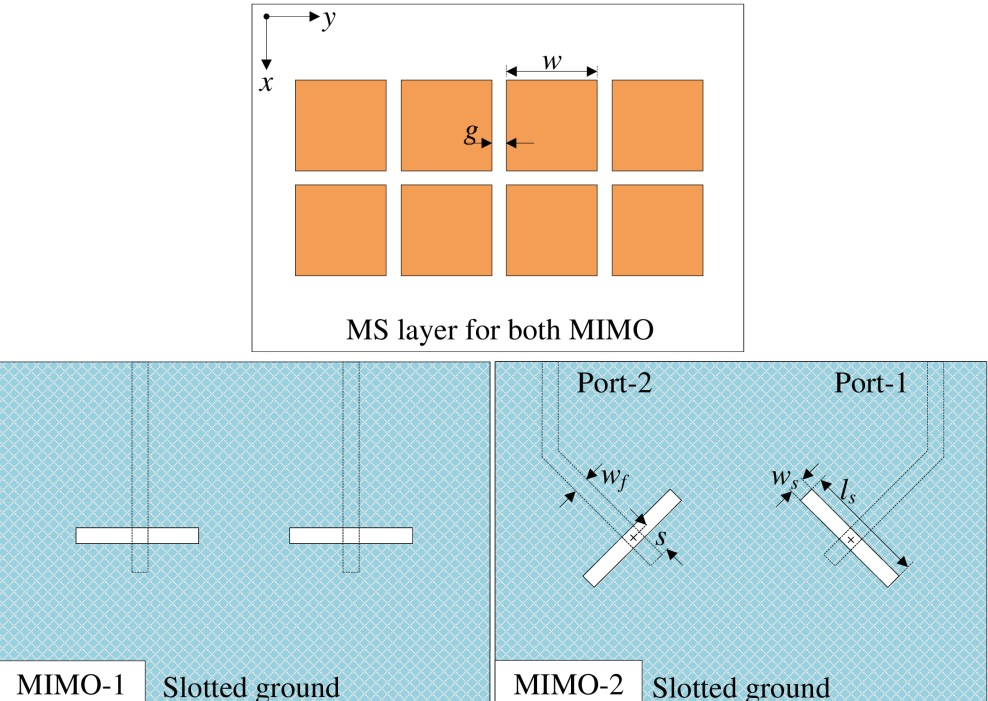

**Fig 3. Geometry of different 2-port adjacent MIMO antennas.**

elements and the MIMO-2 is composed of two Design-2 elements. Note that in both cases, the center-to-center spacings between elements are similar. The optimal dimensions of MIMO-1 are $w = 8.6$, $g = 1.2$, $l_s = 11.2$, $w_s = 1.1$, $s = 2.2$, $w_f = 1.5$ (unit: mm). The optimal dimensions of MIMO-2 are $w = 8.4$, $g = 1.4$, $l_s = 11.2$, $w_s = 1.3$, $s = 3$, $w_f = 1.5$ (unit: mm).

Fig 4 shows the simulated reflection and transmission coefficients of MIMO-1 and -2. It can be seen obviously that the -10 dB impedance BWs are similar in both cases, which is from 4.6 to 6.1 GHz. However, the isolation is significant difference. Strong mutual coupling is observed for MIMO-1 due to the strong coupling happening in the H-plane. On the other hand, as the excited slots are in perpendicular arrangement, the mutual coupling is further suppressed. Here, the isolation is improved from 10 dB for MIMO-1 to about 20 dB for MIMO-2.

## Mutual coupling reduction for opposite MIMO elements

This paper aims to design a 4-port MIMO antenna, which is arranged in $2 \times 2$ configuration. Thus, the decoupling issue between the opposite elements is considered. Fig 5 shows the configuration of the 2-element MIMO antennas as MIMO-3. In this case, two MIMO elements are coupled in the E-plane. Theoretically, the E-plane coupled MIMO antenna results in high mutual coupling. Therefore, two shorting pins are employed to suppress the mutual coupling between these elements. The equivalent circuit of the unit cell with a shorting pin is as a LC resonator. Here, C represents the capacitance between the unit cell and the ground plane, while L is the inductance of the shorting vias. This LC circuit can provide a rejected frequency band, effectively blocking coupling fields from the excited element to the non-excited

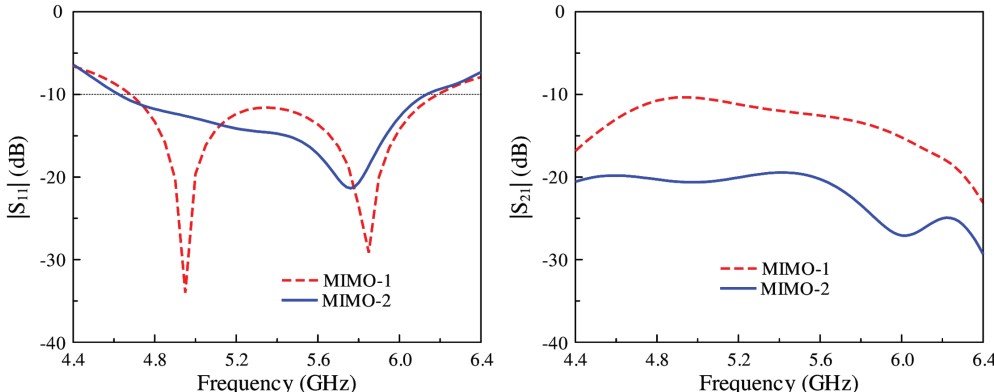

**Fig 4. Simulated $|S_{11}|$ and $|S_{21}|$ of MIMO-1 and MIMO-2 designs.**

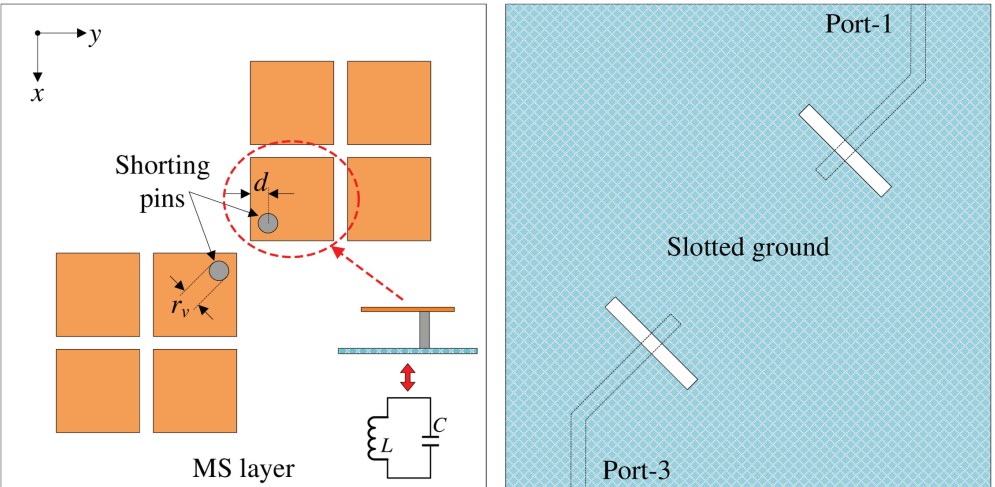

**Fig 5. Geometry of 2-port opposite MIMO antennas with shorting pins.**

one [30]. The optimal dimensions of MIMO-3 are $w = 9.2$, $g = 0.6$, $l_s = 10.5$, $w_s = 1.3$, $s = 3$, $w_f = 1.5$, $r_v = 1.2$, $d = 3.8$ (unit: mm).

The simulated $|S_{11}|$ and $|S_{21}|$ results of MIMO-3 with and without shorting pins are presented in Fig 6. Note that both antennas are optimized so that the element spacing is fixed and the operating frequency range is similar. Obviously, both antennas show good matching performance. However, the isolation of the coupled MIMO is worse than the decoupled MIMO. Here, the isolation is significantly improved from 16 to 28 dB with the presence of the shorting pins. For better understanding, the effect of the shorting pin position on the antenna performance is depicted in Fig 7. When the vias moves closer to the slot, better isolation can be obtained. Meanwhile, the matching performance is quite stable. Fig 8 shows the current distribution on the ground plane at 5.2 GHz for different cases of with and without shorting pins. As seen, strong current is observed around the slot for the first case. On the other hand, it is significantly suppressed with the presence of the shorting pins.

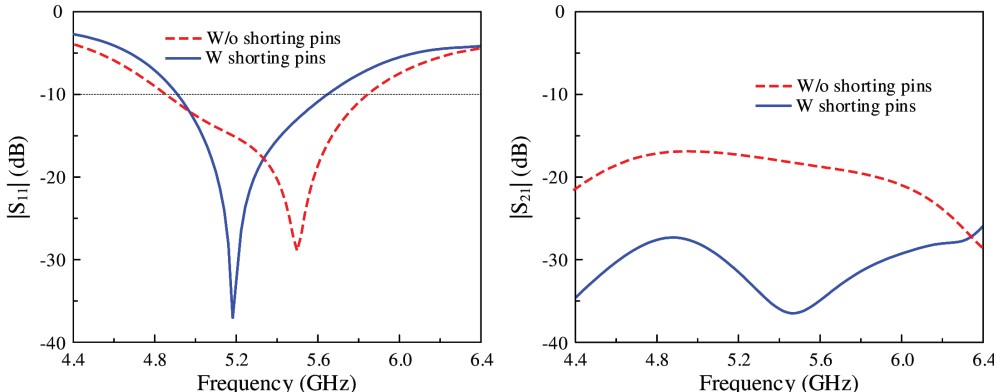

**Fig 6. Simulated $|S_{11}|$ and $|S_{21}|$ of MIMO-3 with and without shorting pins.**

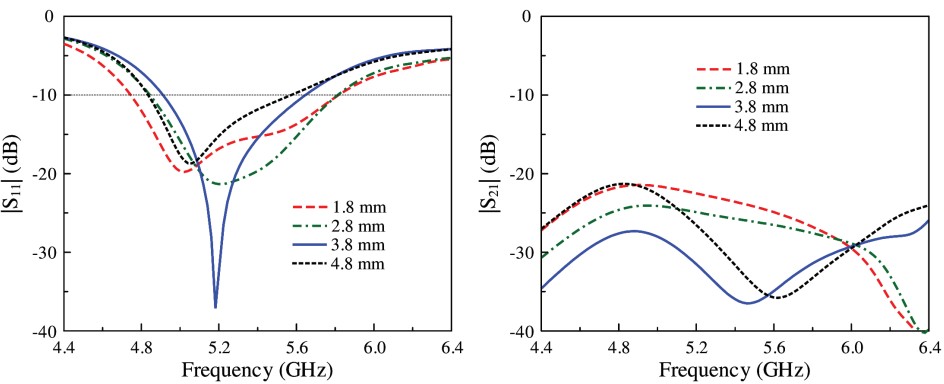

**Fig 7. Simulated $|S_{11}|$ and $|S_{21}|$ of MIMO-3 with different shorting pin position, *d*.**

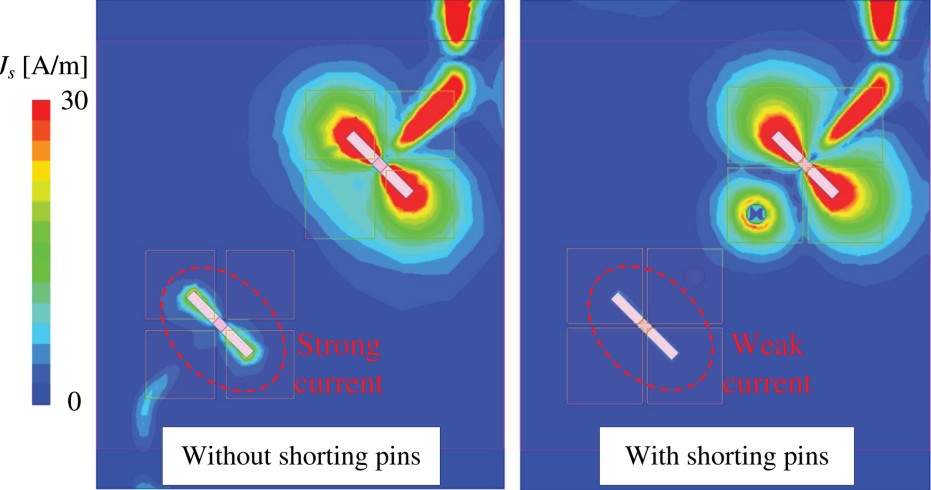

**Fig 8. Simulated current distributions on the ground with and without shorting pins.**

## Final realization of 4-port MIMO antenna

According to the above discussion, the final realization of 4-port MIMO antenna is inspired from operation characteristics of MIMO-2 and MIMO-3. For better understanding the design procedure, Fig 9 shows three different configurations of 4-port MIMO antenna. Design MIMO-4 consists of 4 MIMO elements, which are closely positioned, and no decoupling shorting visa is utilized. For design MIMO-5, four shorting pins are located at the center. Finally, four different shorting pins are additionally introduced in design MIMO-6. The optimal antenna dimensions of MIMO-6 are as follows: $L = 60$, $W = 50$, $w = 9.2$, $g = 0.6$, $l_s = 10.5$, $w_s = 1.3$, $s = 3$, $w_f = 1.5$, $r_v = 1.2$, $d = 1.8$ (unit: mm).

Fig 10 shows the simulated performance in terms of reflection and transmission coefficients of the 4-port MIMO-4. As seen, the isolation values between Port-1 and Port-2 ($S_{21}$) and Port-1 and Port-4 ($S_{41}$) are quite good, which are better than 20 dB. This is consistent with the discussion in the previous section about MIMO-2. Meanwhile, the isolation between the opposite ports, for instance, Port-1 and Port-3 ($S_{31}$), is bad at approximately 10 dB. Next, the radiation pattern of MIMO-4 is considered. As seen in Fig 11, the radiation pattern is

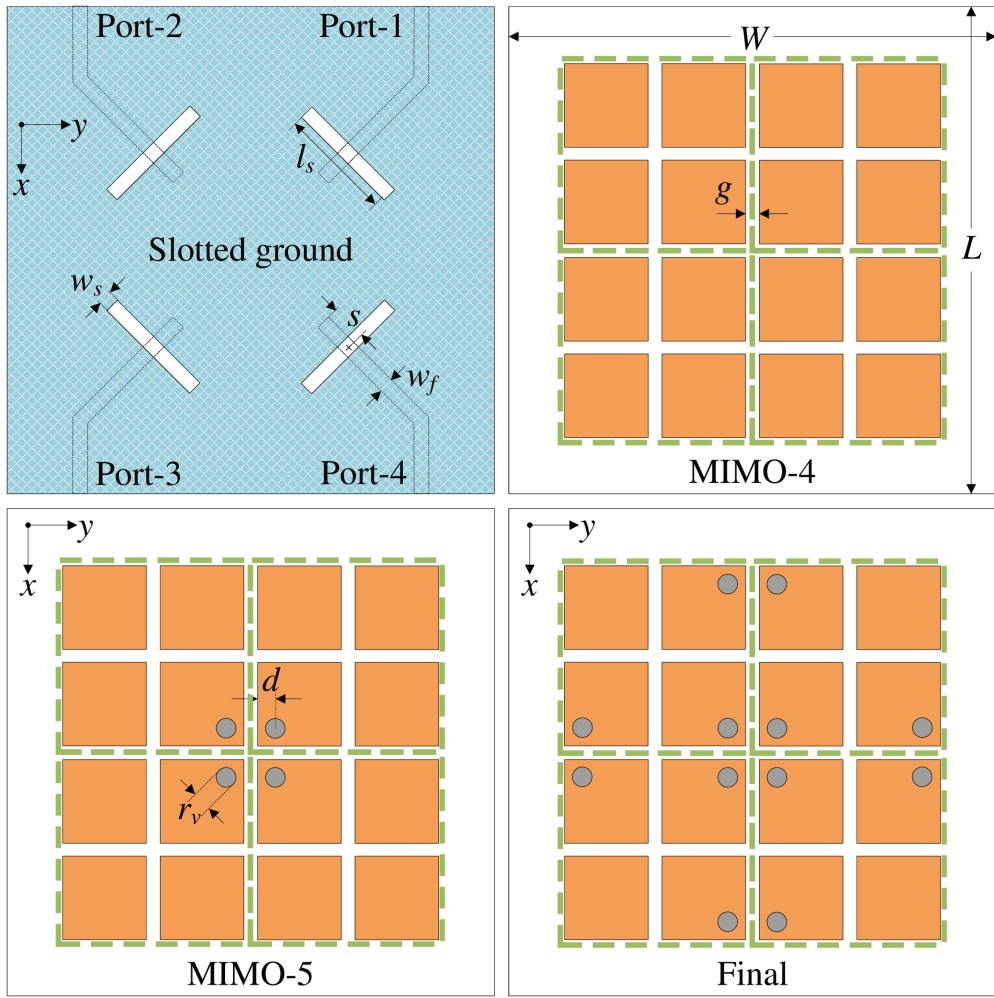

**Fig 9. Geometry of different 4-port MIMO antennas.**

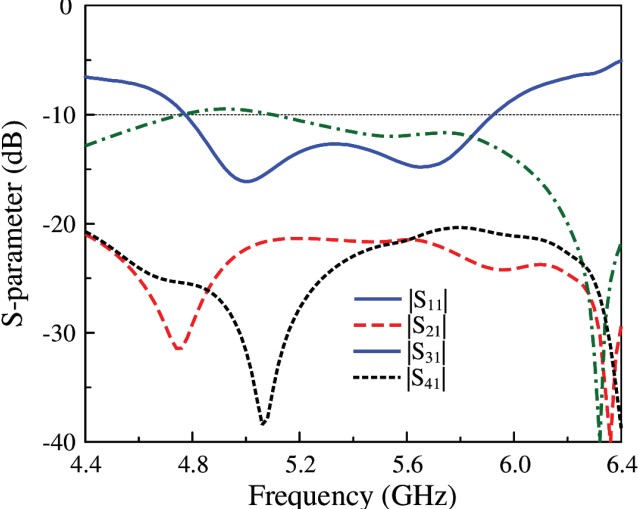

**Fig 10. Simulated S-parameter of MIMO-4.**

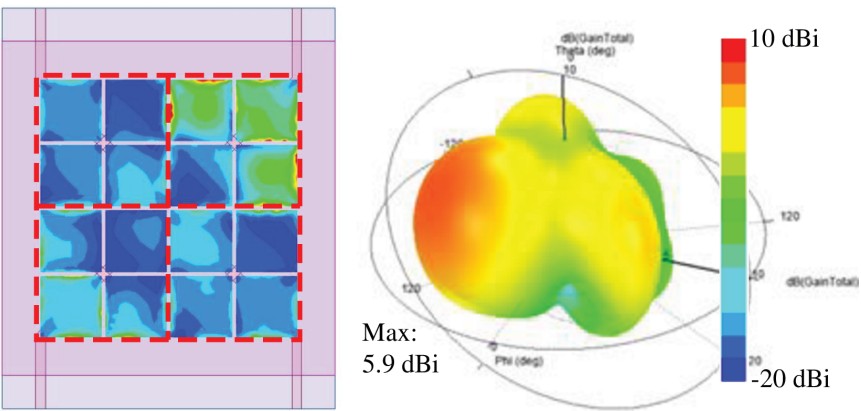

**Fig 11. Simulated 3-D radiation pattern of MIMO-4.**

titled from the broadside direction. This is due to the strong coupling between the MIMO elements, as shown in the current distribution on the MS layer. Four unit cells of the excited MIMO element are excited first. Then, the unit cells of the other MIMO elements are coupled with the excited ones. The phase and magnitude of the coupled unit cells are different. Thus, the main beam is steered off the broadside direction.

Better improvement in isolation can be obtained when four shorting pins are utilized. The simulated performance of the MIMO-5 is presented in Figs 12 and 13. In this case, the isolation between the opposite elements, $S_{31}$, can be improved by about 6 dB. This is consistent with the discussion in the previous section, which explained the effectiveness of using shorting pins to suppress the mutual coupling between the E-plane coupled MIMO elements. Note that the isolation improvement in this case is less than that in the previous section due to the presence of different MIMO elements rather than only two elements. In terms of radiation pattern, the simulated data at 5.2 GHz also demonstrates that the main beam is still not in the broadside direction. A similar phenomenon as the design MIMO-4 is observed due to the

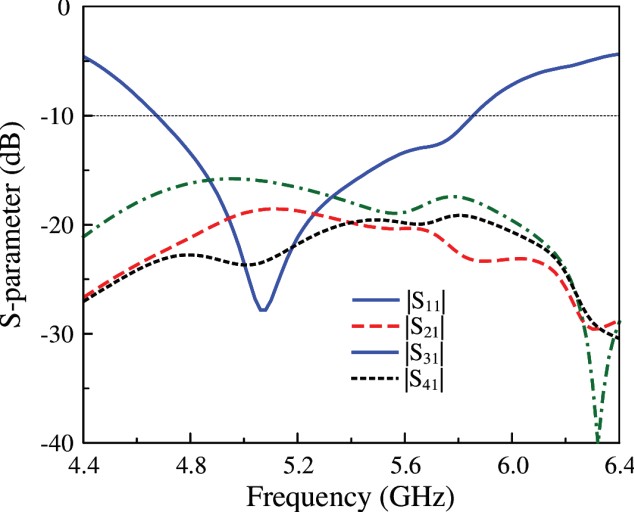

**Fig 12. Simulated S-parameter of MIMO-5.**

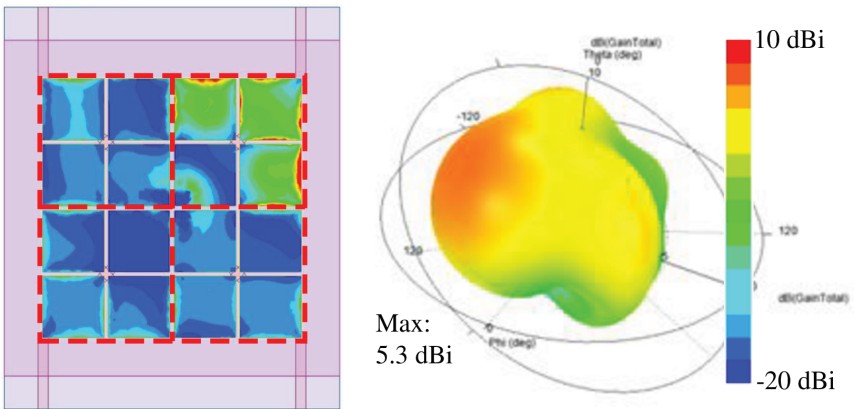

**Fig 13. Simulated 3-D radiation pattern of MIMO-5.**

strong coupling between the MS unit cells of the MIMO elements, as shown in the current distribution on the MS layer.

Further improvements in isolation and radiation pattern can be achieved when more shorting pins are used. As seen in Fig 14, the final 4-port MIMO designated as MIMO-6 has operating bandwidth of 20.5%, ranging from 4.8 to 5.9 GHz. Across this band, the isolation values among all ports are always higher than 20 dB, which is much better than MIMO-4 and MIMO-5 designs. Finally, simulated radiation at 5.2 GHz is considered. The data in Fig 14 indicates that the radiation pattern of MIMO-6 is more broadside than those of MIMO-4 and MIMO-5. This is attributed to the low mutual coupling between the MIMO elements. The radiation from the non-excited unit cell is further suppressed and thus, the broadside beam is obtained.

To validate if the proposed antenna can perform excellently in MIMO systems, the Envelope Correlation Coefficient (ECC) should be considered. ECC is used to evaluate the correlation between a pair of radiating elements regarding the incoming or outgoing signals. The

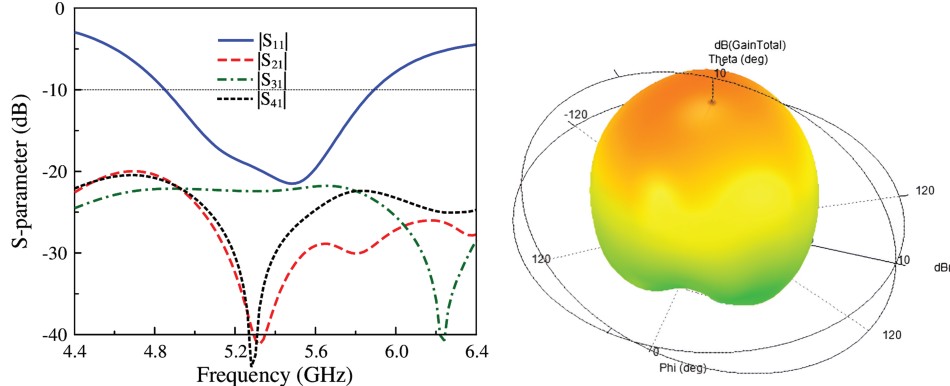

**Fig 14. Simulated S-parameter and 3-D radiation pattern of MIMO-6.**

value of ECC is determined by Eq (1). For the case of 4-port design in this paper, the correlation between port-1 and port-2 could be derived from Eq (1) to Eq (2). By transforming the same way, the correlation between other pairs of ports can also be defined. The calculated ECCs are exhibited in Fig 15. Due to symmetric properties, the calculation is implemented with the data for three pairs of ports, which are ECC12, ECC13, and ECC14. As observed, the correlation values for all the investigated cases remain lower than 0.003 within the operating band, which indicates that the proposed design could offer an outstanding diversity performance.

$$ECC_{ij} = \frac{\left| S_{ii}^* * S_{ij} + S_{ji}^* * S_{jj} \right|^2}{\left( 1 - |S_{ii}|^2 - |S_{ji}|^2 \right)\left( 1 - |S_{jj}|^2 - |S_{ij}|^2 \right)} \tag{1}$$

$$ECC_{12} = \frac{\left| S_{11}^* S_{12} + S_{21}^* S_{22} + S_{13}^* S_{32} + S_{14}^* S_{42} \right|^2}{\left( 1 - |S_{11}|^2 - |S_{21}|^2 - |S_{31}|^2 - |S_{41}|^2 \right)\left( 1 - |S_{12}|^2 - |S_{22}|^2 - |S_{32}|^2 - |S_{42}|^2 \right)} \tag{2}$$

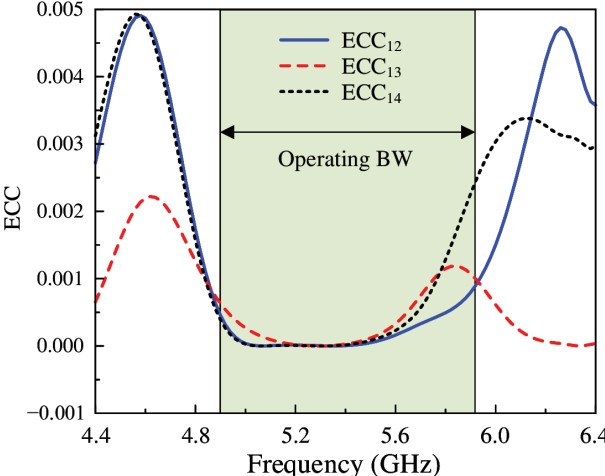

**Fig 15. Calculated ECC values of the proposed 4-port MIMO antenna, MIMO-6.**

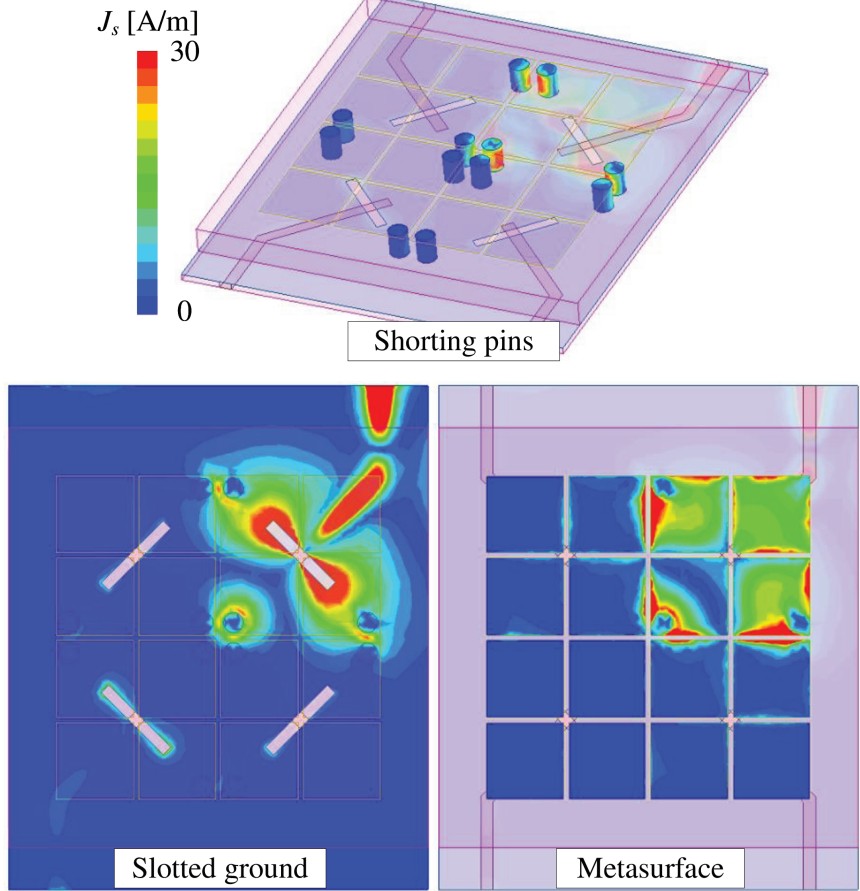

**Fig 16. Simulated current distribution on the final MIMO antenna, MIMO-6.**

For a better demonstration, the simulated current distribution at 5.2 GHz on the final MIMO design, MIMO-6, with Port-1 excitation is investigated. The simulated data on the ground plane, shorting pins, as well as MS layers are respectively depicted in Fig 16. It can be seen obviously that the current is strongly distributed around the slot and 4 unit-cell of the excited MIMO element. On the other hand, the distribution on the other parts of the 4-port MIMO antenna is insignificant. This result further confirms the high isolation mechanism of the proposed antenna. In fact, the mutual coupling between the MIMO elements is suppressed using shorting pins.

## Measured results

The validation of the proposed concept is implemented by experiments on a fabricated antenna prototype, whose photos are shown in Fig 17. Here, the S-parameter is checked with the network analyzer type N5242A, and the far-field patterns are checked with Port-1 excitation in an anechoic chamber. Generally, there is a small discrepancy between the simulations and measurements, which could be attributed to the tolerance of fabrication and measurement setup. Besides, the misalignment in far-field measurement could affect the radiation pattern of the antenna.

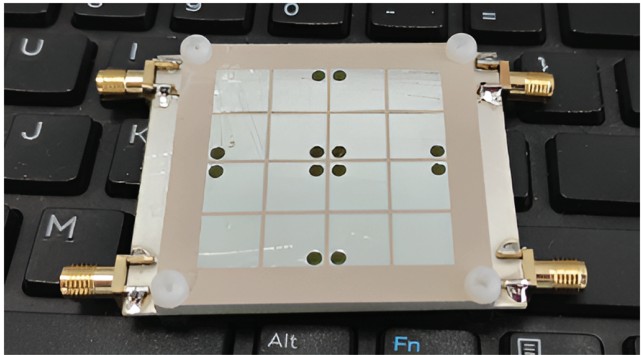

**Fig 17. Photographs of fabricated 4-port MIMO antenna.**

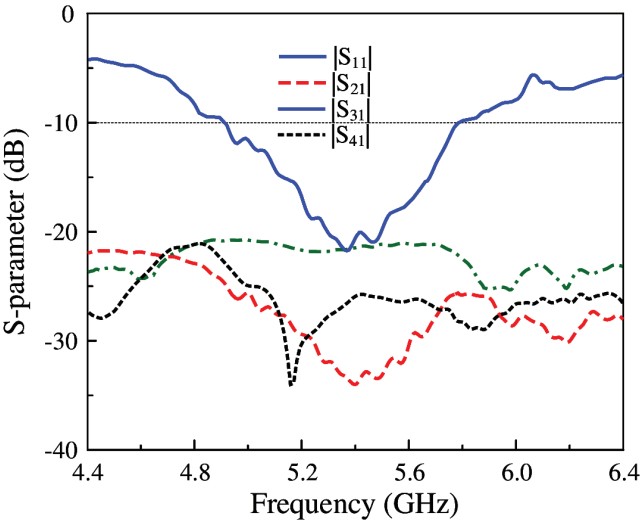

**Fig 18. Measured S-parameter of the proposed 4-port MIMO antenna.**

The simulated and measured reflection coefficient and transmission coefficients between the other ports and Port-1 are illustrated in Fig 18. The measured matching performance with $|S_{11}|$ of less than –10 dB is from 4.9 to 5.8 GHz, equivalent to 16.8%. Meanwhile, the measured isolation values across this band are always higher than 20 dB.

The simulated and measured broadside gains of the proposed MIMO antenna are presented in Fig 19. In the frequency range from 4.9 to 5.6 GHz, the measured broadside gain is always better than 3 dBi. The maximum value of 4.5 dBi can be achieved at 5.2 GHz. Note that in the frequency range of higher than 5.6 GHz, the gain is significantly decreased since the main beam is tilted off the broadside direction. This is due to the unwanted radiation from the coupled unit cells of the other MIMO elements. Additionally, the gain values are low since the utilized low-cost substrate has high loss. It can be improved when better substrates are employed such as Taconic, Roger.

The gain radiation patterns with Port-1 excitation at 5.2 GHz are plotted in Fig 20. Due to the symmetric geometry, the radiation patterns are almost similar for all ports. It can be observed that the proposed antenna radiates quite strongly in the broadside direction. The

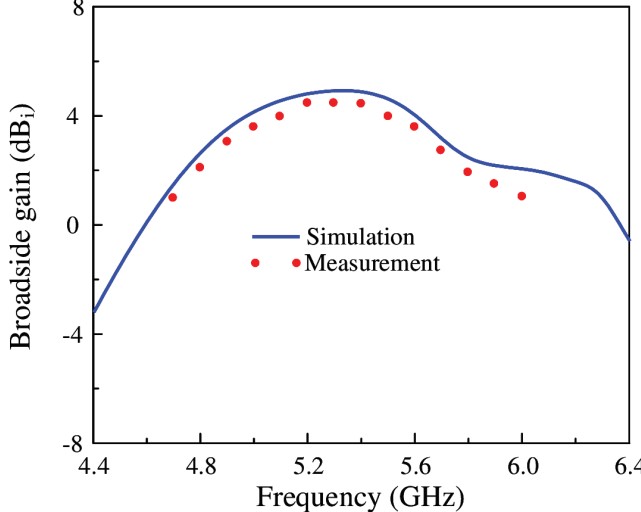

**Fig 19. Simulated and measured broadside gain of the proposed 4-port MIMO antenna.**

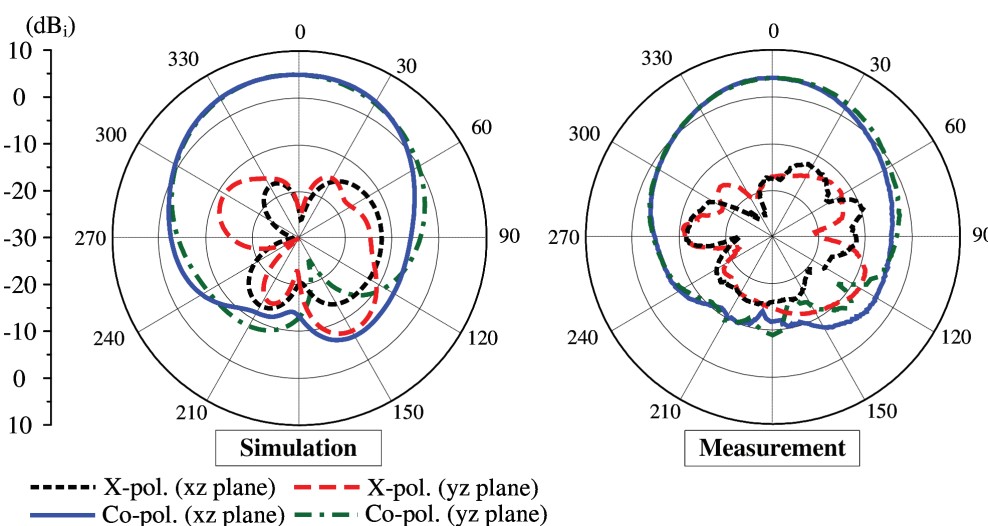

**Fig 20. Simulated and measured radiation patterns of the proposed 4-port MIMO antenna with Port-1 excitation.**

cross-polarization level in the broadside direction is 25 dB lower than the co-polarization level. Meanwhile, the front-to-back ratio is better than 14 dB. In fact, this ratio is not high, and it is a trade-off with the antenna size.

## Performance comparison

To demonstrate the contribution of the proposed work, Table 1 shows the comparison among low-profile and wideband MIMO antennas. It is obvious that despite having small lateral dimensions, the proposed antenna exhibits wide operating BW. Besides, the use of shorting pins inside the MIMO element significantly reduces the center-to-center element spacing of the proposed work while achieving comparable isolation. The design in [29] has smaller

Table 1. Performance comparison among low-profile wideband MIMO antennas.

| Ref. | Antenna type | No. of ports | Overall dimensions ($\lambda$) | Spacing ($\lambda$) (%) | BW (dB) | Min iso. |
|---|---|---|---|---|---|---|
| [22] | DR | 2 | $2.82 \times 2.82 \times 0.40$ | 0.5 | 14.8 | 20 |
| [23] | DR | 2 | $1.52 \times 1.52 \times 0.38$ | 0.5 | 10 | 26 |
| [24] | DR | 2 | $1.90 \times 0.96 \times 0.13$ | 0.5 | 7.3 | 30 |
| [25] | Patch | 4 | $2.00 \times 2.00 \times 0.08$ | 1 | 19.6 | 15 |
| [26] | Patch + PE | 4 | $1.51 \times 1.51 \times 0.04$ | 0.76 | 15.5 | 20 |
| [28] | Patch + MS | 4 | $1.74 \times 1.74 \times 0.04$ | 0.91 | 15.9 | 32 |
| [29] | Slot + MS | 2 | $0.85 \times 0.48 \times 0.03$ | 0.43 | 15 | 25 |
| Prop. | Slot + MS | 4 | $1.08 \times 0.89 \times 0.07$ | 0.35 | 16.8 | 20 |

dimensions; but it is limited to only 2-port. Meanwhile, the DR antennas [22–24] feature high profile configuration and bulky size as well. Finally, it is worth noting that despite having similar MIMO configuration of 4-port, the designs in [25,26,28] have drawbacks of large size and large element spacing. Overall, there is a trade-off between the size and gain of the proposed work. Besides, using low-cost FR4 substrate with high loss tangent is another reason for low gain. The antenna gain can be improved by using low-loss substrates such as Taconic and Roger, but with higher cost. In terms of operating BW, the principal for wideband performance of the presented design is a combination between two resonances, one from the slot and the other from the MS. To improve the operating BW, additional resonances should be produced. It can be accomplished with other shapes of the slot or the unit-cell.

## Conclusion

The 4-port MIMO antenna with wideband operation, high isolation, compact dimensions characteristics is presented and investigated in this paper. The wideband performance is based on a combination of slot and MS resonances. To achieve the MIMO operation, four MIMO elements are arranged in a $2 \times 2$ configuration. The high isolation between the MIMO elements is achieved using shorting pins. The proposed concept has been validated by measurement. The fabricated antenna shows good performance, and it can be used for various wireless applications, including Wi-Fi 5, Wi-Fi 6, and so on.

## Author contributions

**Data curation:** Anh Tran-Tuan.

**Investigation:** Thao Hoang-Thi-Phuong.

**Supervision:** Yem Vu-Van, Hung Tran-Huy.

**Validation:** Thao Hoang-Thi-Phuong.

**Writing – original draft:** Anh Tran-Tuan.

**Writing – review & editing:** Hung Tran-Huy.

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
