## [Decision Letter · Decision Letter 0]

4 Nov 2024

PONE-D-24-38256A simple technique to improve performance of four-port wideband MIMO antenna using shorting pinsPLOS ONE

Dear Dr. Tran-Huy,

Thank you for submitting your manuscript to PLOS ONE. After careful consideration, we feel that it has merit but does not fully meet PLOS ONE’s publication criteria as it currently stands. Therefore, we invite you to submit a revised version of the manuscript that addresses the points raised during the review process.

We look forward to receiving your revised manuscript.

Kind regards,

Maharana Pratap Singh, Ph.D.

Academic Editor

PLOS ONE

**Journal Requirements:**

This work was supported by the Vietnam National Foundation for Science and Technology Development (NAFOSTED) under Grant number 102.04-2023.28

This work was supported by the Vietnam National Foundation for Science and Technology Development (NAFOSTED) under Grant number 102.04-2023.28.

This work was supported by the Vietnam National Foundation for Science and Technology Development (NAFOSTED) under Grant number 102.04-2023.28

Reviewers' comments:

Reviewer's Responses to Questions

**Comments to the Author**

1. Is the manuscript technically sound, and do the data support the conclusions?

Reviewer #1: Yes

Reviewer #2: Yes

2. Has the statistical analysis been performed appropriately and rigorously? 

Reviewer #1: Yes

Reviewer #2: No

3. Have the authors made all data underlying the findings in their manuscript fully available?

Reviewer #1: Yes

Reviewer #2: Yes

4. Is the manuscript presented in an intelligible fashion and written in standard English?

Reviewer #1: Yes

Reviewer #2: Yes

5. Review Comments to the Author

**Reviewer #1:** In this paper, Anh et al. proposed a four-port multiple-input multiple-output (MIMO) antenna with compact size, wideband operation, and high isolation characteristics. The wideband performance is obtained by generating two adjacent resonances, which are respectively produced by a half-wavelength slot and a metasurface (MS). Four MIMO elements are arranged in a 2 × 2 configuration with zero spacing between the MIMO elements to achieve the compact size feature. For mutual coupling reduction, the adjacent elements are positioned so that their polarizations are perpendicular. Meanwhile, the coupling between the opposite elements is suppressed with the aid of shorting pins. The final design has compact size of 1.08 λ × 0.89 λ × 0.07 λ and center-to-center element spacing of 0.35 λ, where λ is the free-space wavelength at 5.2 GHz. The measured operating bandwidth, in which the reflection and transmission coefficients are respectively smaller than –10 and –20 dB, is from 4.9 to 5.8 GHz. Here below are some minor comments:

(1) For the three demonstrated example, the surface current might be added to get a more direct view for the difference between the performances.

(2) Color bar is missed in Fig. 10 and Fig. 11.

(3) The results in Fig. 16 should be compared with the full wave simulation results.

(4) The measured far-field pattern seems a little bit different for the co-pol xz plane. It is better to give a discussion on this point.

(5) The concept of phased tuned metasurfaces might be helpful for the antenna design, for example (a) Nature Communications, 15, 6628, 2024.

**Reviewer #2: **The manuscript presents a solid, incremental contribution to MIMO antenna design, particularly in terms of compactness, isolation, and bandwidth enhancement using shorting pins and a slot/MS combination. While the novelty is modest, it provides practical improvements for modern communication systems. Some additional theoretical depth and discussion of practical applications would enhance the impact of the work.

Please refer to my detailed comments for further improvement. See attachment.

6. PLOS authors have the option to publish the peer review history of their article (what does this mean?). If published, this will include your full peer review and any attached files.

Reviewer #1: No

Reviewer #2: No

---

## [Author Response · Author response to Decision Letter 1]

6 Nov 2024

Original Manuscript ID: PONE-D-24-38256

Original Article Title: “A simple technique to improve performance of four-port wideband MIMO antenna using shorting pins”

To: Reviewer

Re: Response to reviewer

Dear Reviewer,

We sincerely appreciate the time you dedicated to reviewing our paper and for providing such valuable feedback. Your insightful comments have been crucial in enhancing this version of the manuscript. The authors have carefully considered each of your points and have addressed them to the best of our ability.

We are submitting a point-by-point response to your comments, along with an updated manuscript with changes highlighted in red, as well as a clean version without tracked changes.

Best regards,

Reviewer 1: In this paper, Anh et al. proposed a four-port multiple-input multiple-output (MIMO) antenna with compact size, wideband operation, and high isolation characteristics. The wideband performance is obtained by generating two adjacent resonances, which are respectively produced by a half-wavelength slot and a metasurface (MS). Four MIMO elements are arranged in a 2 × 2 configuration with zero spacing between the MIMO elements to achieve the compact size feature. For mutual coupling reduction, the adjacent elements are positioned so that their polarizations are perpendicular. Meanwhile, the coupling between the opposite elements is suppressed with the aid of shorting pins. The final design has compact size of 1.08 λ × 0.89 λ × 0.07 λ and center-to-center element spacing of 0.35 λ, where λ is the free-space wavelength at 5.2 GHz. The measured operating bandwidth, in which the reflection and transmission coefficients are respectively smaller than –10 and –20 dB, is from 4.9 to 5.8 GHz. Here below are some minor comments:

Concern # 1: For the three demonstrated examples, the surface current might be added to get a more direct view for the difference between the performances.

Author response: The author would like to thank the Reviewer for a constructive comment. The surface current graphs for the designs of MIMO-4, MIMO-5, and Final are shown in Fig. 1R. As seen, strong coupling fields are observed on the MS layers of the adjacent elements of MIMO-4 and MIMO-5. Consequently, these designs exhibit high isolation and a significant beam tilt, which are drawbacks of these MIMO antennas (shown in Figs. 10 and 11). On the other hand, the Final design has a small coupling field on the adjacent elements, resulting in higher isolation and more broadside gain.

Fig. 1R. Simulated current distributions on MS layer of different MIMO antennas.

Author action: The current distributions on the MS layers of MIMO-4 and -5 are added to Figs. 10 and 11, respectively. Brief discussions about these distributions are also added to Paragraphs 2, 3, Section “Final realization of 4-port MIMO antenna” of the revised manuscript.

Concern # 2: Color bar is missed in Fig. 10 and Fig. 11.

Author response: Agreed.

Author action: Color bar is added to Fig. 10 and Fig. 11 of the revised manuscript.

Concern # 3: The results in Fig. 16 should be compared with the full wave simulation results.

Author response: As the simulated S-parameter is shown in Fig. 12a, the authors just present the measured results in Fig. 16. Combining these Figures might cause confusion because there are many curves in one figure. Meanwhile, dividing into multiple figures will use lots of space. Thus, the authors believe that showing only measured data is more reasonable.

Concern # 4: The measured far-field pattern seems a little bit different for the co-pol xz plane. It is better to give a discussion on this point.

Author response: In fact, the measured results are affected by the misalignment in measurement setup, and tolerance in fabrication as well. Therefore, a difference between simulation and measurement is unavoidable.

Author action: A brief discussion about the difference between simulation and measurement is added to Paragraph 1, Section “Measured results” of the revised manuscript.

Concern # 5: The concept of phased tuned metasurfaces might be helpful for the antenna design, for example (a) Nature Communications, 15, 6628, 2024.

Author response: According to the suggested reference of Reviewer, the authors have found the following paper https://www.nature.com/articles/s41467-024-50892-y.

However, this aspect does not fall within the scope of the antenna field. Could you please provide us with more details? The authors are willing to response in the next revision stage if possible.

Reviewer 2: The manuscript presents a solid, incremental contribution to MIMO antenna design, particularly in terms of compactness, isolation, and bandwidth enhancement using shorting pins and a slot/MS combination. While the novelty is modest, it provides practical improvements for modern communication systems. Some additional theoretical depth and discussion of practical applications would enhance the impact of the work.

A. Novelty and Unique Contributions:

The manuscript addresses a well-known issue in MIMO antenna design, which is mutual coupling between closely placed antenna elements. The proposed solution of using shorting pins to improve isolation between elements, while maintaining a compact design, brings a modest level of novelty. Specifically:

1. Use of Shorting Pins: Shorting pins as a decoupling mechanism have been used before, but their implementation inside the MIMO elements without requiring additional space is a unique contribution.

2. Combination of Slot and Metasurface (MS) for Wideband Operation: While metasurfaces have been employed for bandwidth enhancement, the integration of slot and MS resonances in a low-profile, compact design is innovative.

3. Compact Design with High Isolation: The design achieves good isolation and bandwidth while maintaining a small footprint (1.08λ × 0.89λ × 0.07λ). This compactness, especially with a center-to-center element spacing of 0.35λ, is another strong point of the work.

However, the novelty is incremental rather than groundbreaking. Many of the techniques discussed, such as orthogonal polarization and metasurfaces for bandwidth enhancement, are well-established in the literature.

Author response: The authors agreed with the Reviewer that the novelty is incremental rather than groundbreaking. Many other decoupling methods have been reported in open literature. However, every method has pros and cons. Noted that several decoupling methods are only suitable for narrow-band antenna. For the proposed work, the use of shorting pins to decouple compact and wideband MIMO antenna is applied to achieve the best performance in comparison to the others listed in Table 1.

B. Content Review:

1. Abstract:

- The abstract succinctly summarizes the proposed design and its key performance metrics. It mentions the compact size, wideband operation, and isolation, but it could benefit from a clearer statement of novelty to make the unique aspects stand out more explicitly.

Author response: Agreed.

Author action: The novelty is succinctly summarized in Abstract of the revised manuscript.

2. Introduction:

- The introduction provides a good overview of the problem (mutual coupling and wideband operation) and discusses previous solutions. It is well- structured, but the novelty of the proposed work could be more clearly emphasized.

- Citations cover a wide range of related work, but it would be beneficial to mention more recent approaches to compact MIMO designs, especially in terms of practical applications for 5G and beyond.

Author response: Agreed.

Author action: The novelty of the proposed work is further emphasized in Paragraph 4, Section “Introduction”. Besides, several recent compact MIMO antennas are added to the revised manuscript as refs [27–29].

3. Technical Soundness:

- Design Methodology: The methodology of integrating slots and metasurfaces is sound, and the simulations support the claimed improvements. However, the design choices, such as the specific placement of shorting pins, could benefit from a deeper theoretical justification beyond the observed simulation results.

Author response: The authors would like to thank the Reviewer for a very constructive comment. Indeed, a deeper theoretical justification on the position of shorting pins is more beneficial than observing simulated data. Fig. 2R shows the equivalent circuit of the unit cell with a shorting pin, where the presence of the vias creates an LC resonator. Here, C represents the capacitance between the unit cell and the ground plane, while L is the inductance of the shorting vias. This LC circuit can provide a rejected frequency band, effectively blocking coupling fields [1R]. With respect to the position of this rejected band circuit, it has been optimized based on simulation.

Fig. 2R. Equivalent circuit of the unit cell with shorting vias.

[1R] C. L. Wang, G. H. Shiue, W. D. Guo, and R. B. Wu, “A systematic design to suppress wideband ground bounce noise in high-speed circuits by electromagnetic-bandgap-enhanced split powers,” IEEE Trans. Microw. Theory Tech., vol. 54, no. 12, pp. 4209–4217, Dec. 2006.

Author action: The equivalent circuit of the utilized decoupling network is added to Fig. 5. Further discussion is also added to Paragraph 2, Section “Mutual coupling reduction for opposite MIMO elements” of the revised manuscript.

- Figures and Diagrams: Figures (e.g., Fig. 1, 2, 5, 6) provide clear visual representations of the design and performance. The geometry and current distribution figures are particularly useful for understanding the design’s effectiveness.

The radiation patterns (Fig. 10 and Fig. 18) show that while broadside radiation is achieved, there is some beam tilt. This could be discussed more thoroughly, especially regarding the practical implications for real-world usage.

Author response: The authors agree with the Reviewer that the current distribution figures will be useful for understanding the designs’ effectiveness. The final goal of the paper is to design a 4-port MIMO antenna; thus, the current distribution of MIMO-4, -5 are included.

Regarding the tilted beam, this is due to the unwanted radiation from the MS of adjacent MIMO elements. Although using the shorting pins can suppress the coupling fields from the excited element to the non-excited elements, there always exists small amount of coupling fields on the non-excited elements. As a result, this leads to some degree of beam tilt.

Author action: In accordance with the Reviewer’s comment, the current distributions of MIMO-4, -5 are added to the revised manuscript along with brief discussions about them. Besides, an explanation about the beam tilt is added to Paragraph 4, Section “Final realization of 4-port MIMO antenna”.

4. Results:

- Bandwidth and Isolation: The achieved bandwidth of 4.9 to 5.8 GHz (16.8%) with isolation above 20 dB is strong. However, the bandwidth is somewhat narrow compared to the latest wideband MIMO antennas. Is there any potential to further enhance this? Discussing this limitation in more detail would add depth.

- Gain: The maximum broadside gain of 4.5 dBi is reasonable for the proposed size, but it is not particularly high. The trade-off between gain and size could be more explicitly discussed.

- Measurement vs. Simulation: The manuscript acknowledges slight discrepancies between simulated and measured results (Fig. 16 and 17), which is typical in such designs. However, further explanation of the causes (e.g., substrate tolerance, fabrication imperfections) would strengthen the argument.

Author response: The authors agree with the Reviewer that there is a trade-off between the overall size and gain of the proposed work. Besides, using low-cost FR4 substrate with high loss tangent is another reason. The antenna gain can be improved by using low-loss substrates such as Taconic and Roger, but with higher cost. In terms of operating BW, the principal for wideband performance of the presented design is a combination between two resonances, one from the slot and the other from the MS. To improve the operating BW, additional resonances should be produced. It can be accomplished with other shapes of the slot or the unit-cell.

Finally, as the Reviewer points out, the discrepancies between simulation and measurement could be attributed to the fabrication tolerance and the imperfection in measurement setup.

Author action: A brief discussion about the performance of the antenna is added to Paragraph 1, Section “Performance comparison”. Additionally, the reasons for discrepancies between simulation and measurement are briefly mentioned in Paragraph 1, Section “Measured results” of the revised manuscript.

5. Novel Contributions Section:

- A dedicated section or clearer presentation of the unique aspects of the design would help the reader quickly grasp the novel elements, especially in comparison with existing solutions. This is partially covered in the performance comparison table (Table 1), but more emphasis on the compactness, isolation, and slot/MS combination would add clarity.

6. Tables:

- Table 1: The performance comparison table is well-constructed, showing that the proposed antenna has competitive characteristics in terms of size, bandwidth, and isolation. It demonstrates that despite its compact size, the proposed design achieves similar or better performance compared to larger antennas.

Author response: Agreed.

Author action: More emphasis on the antenna characteristics is added to Paragraph 1, Section “Performance comparison” of the revised manuscript.

C. Strengths:

1. Compact Design: The small footprint and low profile make this antenna suitable for space-constrained applications, such as mobile devices or compact IoT systems.

2. High Isolation: The shorting pin technique effectively reduces mutual coupling, as shown by the isolation improvement in the S-parameters.

3. Simulation Support: The manuscript is well-supported by simulation and measurement results, which align closely in most cases.

Author response: The authors would like to thank the Reviewer for your valuable time spent on understanding our work. The summaries about the strengths of the presented work are very useful for us to emphasize them throughout the manuscript.

D. Areas for Improvement:

1. Theoretical Analysis: The manuscript would benefit from a deeper theoretical discussion of how the shorting pins and slot/MS combination work together to enhance bandwidth and isolation.

Author response: Agreed.

Author action: The wideband principal is further discussed in Paragraph 2, Section “Wideband MIMO element”. Besides, the function of the shorting pins in isolation enhancement is added to Paragraph 2, Section “Mutual coupling reduction for opposite MIMO elements” of the revised manuscript.

2. Practical Applications: While the manuscript is technically sound, it could be strengthened by discussing practical implications, such as real-world deployment in 5G or Wi-Fi 6E systems. How would this design perform under different environmental conditions or in a densely populated MIMO system?

Author response: The authors agree with the Reviewer that to demonstrate the usefulness of the proposed design in practical applications, measurement under practical communication systems is necessary. However, the proposed design is just tested in an anechoic chamber. The suggestions from the Reviewer will be our work in the future.

3. Beam Steering: The tilt of the radiation pattern is noted but not deeply analyzed. Could this be mitigated with further design modifications?

Author response: The tilt of the radiation pattern is caused by the radiation from the non-excited MIMO elements. In fact, this can be mitigated by suppressing the coupling fields on these elements. Increasing the element-to-element spacing could be effective, but there is a trade-off with the overall size of the antenna. Currently, we are working on the same topic and focusing on solving this problem. Again, the authors would like to thank the Reviewer for many constructive comments on our manuscript.

Author action: The explanation about the tilt of the radiation pattern is added to Paragraph 4, Section “Final realization of 4-port MIMO antenna”.

E. Conclusion:

The manuscript presen

---

## [Decision Letter · Decision Letter 1]

10 Mar 2025

A simple technique to improve performance of four-port wideband MIMO antenna using shorting pins

PONE-D-24-38256R1

Dear Dr. Tran-Huy,

We’re pleased to inform you that your manuscript has been judged scientifically suitable for publication and will be formally accepted for publication once it meets all outstanding technical requirements.

Kind regards,

Sachin Kumar, Ph.D.

Academic Editor

PLOS ONE

Additional Editor Comments (optional):

The authors have carefully addressed the reviewers' comments, and the manuscript can accepted for publication.

Reviewers' comments:

Reviewer's Responses to Questions

**Comments to the Author**

1. If the authors have adequately addressed your comments raised in a previous round of review and you feel that this manuscript is now acceptable for publication, you may indicate that here to bypass the “Comments to the Author” section, enter your conflict of interest statement in the “Confidential to Editor” section, and submit your "Accept" recommendation.

Reviewer #1: All comments have been addressed

2. Is the manuscript technically sound, and do the data support the conclusions?

Reviewer #1: Yes

3. Has the statistical analysis been performed appropriately and rigorously? 

Reviewer #1: Yes

4. Have the authors made all data underlying the findings in their manuscript fully available?

Reviewer #1: Yes

5. Is the manuscript presented in an intelligible fashion and written in standard English?

Reviewer #1: Yes

6. Review Comments to the Author

Reviewer #1: (No Response)

7. PLOS authors have the option to publish the peer review history of their article (what does this mean?). If published, this will include your full peer review and any attached files.

Reviewer #1: No

---

## [Editor Report · Acceptance letter]

PONE-D-24-38256R1

PLOS ONE

Dear Dr. Tran-Huy,

I'm pleased to inform you that your manuscript has been deemed suitable for publication in PLOS ONE. Congratulations! Your manuscript is now being handed over to our production team.

Kind regards,

on behalf of

Dr. Sachin Kumar

Academic Editor

PLOS ONE